# Facile Incorporation of Carbon Nanotubes into the Concrete Matrix Using Lignosulfonate Surfactants

**DOI:** 10.3390/ma17204972

**Published:** 2024-10-11

**Authors:** Aleksandra Kostrzanowska-Siedlarz, Krzysztof Musioł, Tomasz Ponikiewski, Dawid Janas, Marian Kampik

**Affiliations:** 1Faculty of Civil Engineering, Silesian University of Technology, Akademicka 5, 44-100 Gliwice, Poland; 2Faculty of Electrical Engineering, Silesian University of Technology, Akademicka 10, 44-100 Gliwice, Poland; 3Faculty of Chemistry, Silesian University of Technology, B. Krzywoustego 4, 44-100 Gliwice, Poland

**Keywords:** multi-walled carbon nanotubes, MWCNTs, concrete, plasticizers, electrical impedance, smart composites, nanoconcrete

## Abstract

One of the ways to turn concrete into smart concrete involves the incorporation of conductive fillers. These fillers should be evenly distributed in the matrix to enable the charge propagation necessary for sensing. To homogenize the mixture, typical surface-active chemical compounds are routinely employed. Unfortunately, their presence often negatively impacts the characteristics of concrete. In this work, we show that conductive multi-walled carbon nanotubes (MWCNTs) can be included in the concrete matrix by using off-the-shelf lignosulfonate-based plasticizers. These plasticizers showed a much-improved capability to disperse MWCNTs compared to other routinely used surfactants. They also prevented a significant deterioration of the consistency of the mixture and inhibited the acceleration of the hydration process by MWCNTs. In concretes with MWCNTs and lignosulfonate-based plasticizers, the mechanical properties were largely preserved, while the nanocomposite became electrically conductive. Consequently, it enabled evaluation of the condition of the material by electrical impedance measurements.

## 1. Introduction

The ever-increasing technological expectations of modern civilization put more and more burden on scientists. There is a perpetual need to design advanced materials that would not only sustain the current pace of progress but preferentially accelerate it. One of the most promising fields of research that is able to keep up with these requirements deals with so-called nanomaterials [1,2,3]. They are widely applied in a growing number of R&D areas, such as medicine [4,5], electronics [6], or civil engineering [7,8,9,10,11]. Nevertheless, their utility for civil engineering remains to be unleashed to a large extent. One of the prominent areas of growing attention concerns the development of self-sensing concrete [12,13,14,15,16,17]. Electrically conductive nanosensor materials such as nanocarbon are introduced to gauge stress or deformations of structures. Besides damage tracking, the application of such a functional material enables one to monitor traffic on the roads [13,17,18,19,20,21,22,23,24,25]. Nanoconcrete adds a spectrum of new properties to the regular concrete that has been used for hundreds of years, such as the ability to self-heal [26,27]. The particle shape/size, specific surface, the concentration of dispersed phase, etc., have a significant impact, so the type of nanofiller and its amount have to be optimized for an envisioned application [28].

Carbon nanotubes (CNTs) are an example of a promising functional additive, the incorporation of which has a multidimensional impact on the properties of the resulting concrete composite. CNTs seal the microstructure by reducing total porosity, accelerating the hydration of calcium silicate (CSH) and gel formation, and also contributing to the improvement of Young’s modulus [29,30,31]. The tensile strength of CNTs and the modulus of elasticity are about 500 and 20 times higher than that of steel, respectively. Therefore, the inclusion of nanocarbon into concrete can indisputably enhance the mechanical properties of the resulting material [32,33,34,35]. Even a small amount of such a modifier at the level of just 0.08% was found sufficient to observe a notable improvement [36,37,38,39,40,41]. Furthermore, the literature reports an approximately 18–25% rise in the compressive strength of nanoconcrete upon the insertion of CNTs [42,43]. The improvement of the operational properties of concrete depends on the type of nanofiller.

However, it has to be taken into account that, for the nanomodification to be successful, effective dispersion of the filler in the matrix is necessary [44,45,46,47]. Appropriate deagglomeration of nanoparticles is needed to reach high performance because it ensures strong interaction between the components. CNTs are a somewhat problematic filler despite their merits. This material is notably hydrophobic and has a very high tendency to form aggregates due to strong van der Waals forces [48,49]. As a consequence, it is hard to disperse it in almost any medium, so clumps or bundles frequently develop in the material [44]. Because of the unfavorable physicochemistry of CNTs, the influence of these factors may give rise to the formation of weak zones. In such a case, instead of increased strength, one is faced with the deterioration of the overall mechanical properties of concrete due to poor stress transfer [50,51,52,53]. There are various methods that may be used in combination to alleviate these unwanted characteristics of nanomaterials [45,46,54,55]. They can be divided into three basic categories: mechanical, chemical, and physical. Mechanical methods, such as shear mixing, ball milling, or sonication [21], all engage various forms of energy to individualize nanomaterials. However, such vigorous conditions also have a destructive impact on the microstructure of the filler if excess power is used for the process. For example, prolonged sonication at high power has a documented deterioration action on the CNTs by cutting them and introducing defects into the structure [56,57]. These issues should be avoided as the high purity of the CNT filler is detrimental to enhancing the properties of the matrix. Optimum conditions have to be employed to ensure a balance between making good dispersion and causing collateral damage [58]. Another way of improving the compatibility of CNTs with concrete is chemical modification. In such an approach, chemical groups of a hydrophilic character (hydroxyl, carboxyl, etc.) are grafted onto the CNT surface by a range of oxidation methods [47]. The processing very much improves the wettability of nanocarbon. However, the modification has to be conducted to the smallest possible extent in this case as well. Aggressive chemical functionalization, using strong acids and high temperatures, can bring about structural defects resulting in inferior properties of the fillers. These commonly manifest as a reduction in strength, electrical conductivity, and so forth [59,60].

Lastly, the most gentle technique to make nanomaterials compatible with concrete involves physical modification of the surface. The addition of appropriate surface-active compounds, which deposit onto nanomaterials and mediate the interaction between the hydrophobic CNT surface and the hydrophilic medium. The hydrophobic nature of CNTs can be cloaked upon proper encapsulation with surfactant species to make them dispersible [47,61]. There is a large number of chemical compounds that can be used for such purposes, like Triton X-100, sodium dodecylbenzenesulfonate, and sodium cholate. Unfortunately, the use of such chemical surfactants is often a burden, as they can deteriorate the properties of the matrix themselves as they act as a barrier to charge or stress transfer [62,63]. Liquidizing and plasticizing admixtures are a group of chemical compounds commonly used to improve the rheological and other properties of the cement matrix. Hence, their presence is welcome (in contrast to the inclusion of typical surfactants mentioned above) [64,65,66]. The aim of this research was to evaluate various surfactants as liquidizing and plasticizing admixtures in order to successfully disperse CNTs for their subsequent use as an admixture in mortars, concrete mixtures, and concretes. For the reason that, commonly, the tests on MWCNTs are only performed on cement pastes [67,68,69], we expanded the scope of experiments in order to accurately recognize the action of MWCNT admixtures in the phase system with the aggregate. To prove the suitability of MWCNTs, we tested the consistency, setting time, compressive strength, and bending strength, and, consequently, the target performance of the suspension with MWCNT in concrete by measuring electrical impedance. The broad spectrum of experiments conducted revealed considerable insight into the use of nanocomposites in civil engineering.

## 2. Experimental

### 2.1. Materials

Portland cement CEM I 42.5 R was used as the base for mortar and concrete produced in this study. The composition and properties of cement are given in Table 1 and Table 2, respectively.

Various admixtures at different concentrations have been assessed. Five types of plasticizers and superplasticizers obtained from commercial sources were tested as admixtures (Table 3). The aim was to find a chemical compound that would have a dual impact on the properties of the material. Firstly, it needs to reduce the amount of mixing water to improve the workability of cement. Secondly, it should also have the ability to disperse nanocarbon to avoid the inclusion of typical surfactants, the use of which could negatively impact the properties of the resulting concrete composites.

Multi-walled carbon nanotubes (MWCNTs) of technical grade (NC7000, Nanocyl, Sambreville, Belgium) were used as a filler. According to the manufacturer, their average diameter and length were 10 nm and 1.5 µm, respectively. The material had a surface area of between 250 and 300 m^2^/g and a volume resistivity of 10^−6^ Ω·cm. Standard sand in line with PN EN 196-1 [70] was used to make the mortars. The aggregates for concrete used in this study consisted of natural sand (0–2 mm) and gravel aggregate fractions (2–4 mm, 4–8 mm, and 8–16 mm having a density of 2.62, 2.74, and 2.74 kg/m^3^, respectively). The grading curves of the aggregate are presented in Figure 1.

### 2.2. Methods

#### 2.2.1. Preparation of MWCNT Dispersion

The nanocarbon material was ultrasonicated (Hielscher UP200St) in the presence of various concrete admixtures (Table 3) to prepare aqueous MWCNTs dispersions. It was necessary to modulate the power and sonication time to ensure the generation of well-dispersed material. The compositions of individual mixtures and the processing parameters are summarized in Table 4. Cooling by keeping the samples in an ice bath was necessary to avoid the rise in temperature, which would deteriorate the quality of the dispersion. Visual inspection was used to assess the quality of the prepared suspensions. The main emphasis was on the color and transparency of the suspension and its behavior at the edge of the liquid and on the walls of the container. Several sonication attempts with different sonication times were taken to find processing conditions leading to the generation of homogeneous MWCNT dispersions. Suspensions that exhibited satisfactory quality are marked in gray in Table 4.

#### 2.2.2. Characterization of MWCNTs

Raman spectra were acquired from 1225 to 1700 cm^−1^ to quantify the quality of the employed MWCNTs (inVia Renishaw, λ = 633 nm, 0.05% total laser power). An average I_D_/I_G_ ratio, which is a common measure of the purity of the material, was calculated along with the corresponding standard deviation. The sample was characterized in multiple locations and using prolonged laser illumination to ensure the statistical significance of the recorded data and a good signal-to-noise ratio.

#### 2.2.3. Preparation of Mortars and Fresh Concrete for Analysis

A mixture of MWCNTs (1.2 g), water (250 g), and a plasticizer, PL1 (10 g), was used to prepare nanoconcretes for analysis. The compositions of mortars and concrete based on this combination are shown in Table 5 and Table 6, respectively. The fresh concrete and mortars were prepared in planetary mixers of the capacity of 30 dm^3^ and 2 dm^3^, respectively, using analogous mixing procedures. Portland cement, sand, and gravel were dry-mixed in the mixer for 30 s. Next, water containing a suspension with PL and MWCNT was added. The total mixing time was 5 min.

#### 2.2.4. Evaluation of Mortar Consistency

The consistency test was executed in line with the standard PN-EN 1015-3: 2000 [71]. The experiment began with placing the mold in the center of the table disk, into which the prepared mortar was introduced in two layers. Each of them was compacted with 10 impacts of the rammer. The excess mortar was removed with a flat scraper, and the dirt on the table disk was wiped. After 15 s, the mold was slowly removed by lifting, and the shaking machine was turned on. The diameter of the resulting mortar sample was measured with a caliper in two directions perpendicular to each other.

#### 2.2.5. Testing the Setting Time of Mortar

The samples of standard-specific mortars were prepared for testing. The consistency was determined based on the standard PN-EN 1015-4: 2000 [72] using the penetrometer method. First, the mold with a diameter of 80 mm and a height of 70 mm was filled with two layers. Each of them was compacted by hitting it with a rammer 10 times. The upper surface was leveled, and the excess mortar was collected. Then, the penetrometer plunger was set 100 mm above the top surface of the sample, and the hook was released, allowing free fall. It was determined that each batch should have a consistency corresponding to the plunger’s penetration depth of 3.5 cm to reach an appropriate setting time in the subsequent parts of this study.

When analyzing the impact of MWCNTs on the consistency of the mortar, an increased amount of water with the increase in the applied amount of nanomaterial was taken into account. The batches prepared to test the time of the mortar setting were in line with the standard PN-EN 480-2:2004 [73]. For that purpose, the Vicat apparatus was used. A ring in the shape of a truncated cone of a depth of 40 mm with an inner upper diameter of 70 mm and a lower diameter of 80 mm placed on a glass base was filled with mortar immediately after mixing. The top surface was leveled so that it was smooth, and then, directly, the entire sample was transferred to a steel container with water. In this way, the upper surface was covered with water. Thus, the conditions set in the standard were met, i.e., the sample should be stored in a chamber with a relative humidity of not less than 90%. The prepared mortar was placed in the Vicat apparatus. The time of the first test was set on the device at 180 min, counted from the end of mortar mixing. The experiment consisted of dipping the needle of the machine vertically into the mortar. The reading was performed when the needle stopped dipping or after 30 s from the release of the rammer, depending on which of the above occurred faster. We recorded the reading, indicating the distance from the tip of the needle to the base plate and the time elapsed from the end of the mortar mixing process. The dip measurements were repeated every 15 min. The analysis was stopped when the distance between the needle and the glass base was 4.0 mm, which is equivalent to the beginning of the mortar setting.

#### 2.2.6. Testing the Consistency of the Concrete Composite

The consistency of the concrete composite was tested in compliance with the standard PN-EN 12350-5 [74] using the flow table test. The method is reliable for flow values from 340 mm to 600 mm. The experiment consisted of placing the cone mold, moistened from the middle, centrally on the flow table. The mold prepared this way was filled with concrete composite in two layers, each compacted by (lightly) hitting 10 times with a wooden rammer. The excess mix protruding above the upper edge of the mold was removed. Then, the mold was raised after 30 s. This operation was carried out within approx. 3–6 s. After this operation, 15 cycles were performed, consisting of lifting the top table plate and its free fall. Then, the spread of the mixture in two perpendicular directions was measured. The results were averaged and rounded to 1 mm.

#### 2.2.7. The Methodology of Strength Evaluation

The strength test involving flexion and compression of the hardened mortar was carried out in line with the standard PN-EN 1015-11:2001 [75] after 28 days and additionally after 1 and 7 days (Figure 2). The strength test involving compression of the hardened concrete was carried out in line with the standard PN-EN 12390-1:2013-03 [76] after 7 and 28 days.

##### Evaluation of the Flexural Strength of Mortar

For flexural tests, prism-shaped samples of mortar with dimensions of 160 mm × 40 mm × 40 mm were made using metal molds consisting of movable walls that formed three chambers after assembly. From each type of mortar, three molds were made containing three beams. The beams were unmolded and weighed after 1 day, then placed in a climatic chamber until the moment of testing. Flexural tests were performed for three beams of the same composition for each composition (Z0, Z1, Z2, Z3, Z4) and for each testing time. The results given in Table 7 are averages of three results obtained for a specific composition. If one of the results is 5% greater than the calculated average, the result is incorrect. In this case, it should be discarded, and the average of the remaining results calculated. The test should be repeated when two results differ from the calculated mean by more than 5%. To determine the flexural strength, the bars are placed on cylindrical supports with a diameter of 10 mm and a span of 100 mm, and the sample is loaded with a centrally located cylinder of the same length and diameter as the supports. The specimen undergoing flexural strength testing is shown in Figure 2b. The bars were loaded to the point of breaking. The concentrated force acted in the middle of the span.

##### Evaluation of the Compressive Strength of Mortar and Concrete

The compressive strength test of the mortar was carried out on the halves of the bars obtained from the flexural tests.

Concrete samples were formed in accordance with the PN-EN 12390-1:2013-03 standard. Three samples were made for each test date. The samples were molded in plastic molds. Samples with dimensions of 10 × 10 × 10 cm were made. The samples were disassembled after 1 day and were stored in a climate chamber with water at a temperature of 20 ± 2 °C until testing. The strength test was carried out using the testing machine (Figure 2c). The load was applied perpendicular to the forming direction. The speed of the applied load was 0.6 ± 0.2 MPa/s.

The destructive force is transmitted using metal washers measuring 40 mm × 40 mm and 10 mm thick. The top plate is fitted with holes for guide pins. It is supported by springs. In the center of each plate, there is a rectangular plate. They are placed in such a way that, when the upper plate is pressed against the lower plate, the surfaces overlap. The halves of the bars, which are obtained after bending, are placed symmetrically on the surface of the plate. The pressure increase per sample should be 15–20 kg/cm^2^ per second. When crushing the sample, it is important to read the pressure on the pressure gauge. The measure of compressive strength is nothing else than the ratio of the force indicated by the dynamometer or corresponding to the greatest pressure indicated by the manometer to the surface area. The compressive strength of the mortar is the arithmetic average of the strength of the six halves. If one of the results is 5% greater than the calculated average, the result is incorrect. It should be discarded, and the average of the remaining results should be calculated. The test should be repeated when two results differ from the calculated mean by more than 5%.

#### 2.2.8. Measurements of the Electrical Impedance

Ten concrete cuboidal specimens (five references and five with MWCNTs), each with dimensions of 10 cm prepared as described in section above, were tested. The specimens were placed for 28 days in a curing tank stored in a temperature-controlled laboratory environment (22 ± 1 °C). To ensure proper connection between the electrode and the sample, gold-plated electrodes with high surface flatness were used. Additionally, a conductive gel (Unidem) was placed at the plate–specimen interface. A wooden measuring adapter with a pneumatic supply and pressure reduction was prepared to obtain a repeatable pressure of the electrodes on the tested specimen (Figure 3). A Keysight E4980A LCR meter (Figure 3) was employed to obtain the impedance of the concrete samples over the frequency range of 10 Hz–2 MHz. The impedance was recorded at 50 frequency points in voltage drive mode, with a relatively low signal amplitude of 1 V to reduce lead inductive effects at the higher frequencies. A Visual Basic for Application (VBA) script was prepared to automate the measurement process. It took approximately 1 min to complete one measurement sweep. All measurements were undertaken in a temperature-controlled laboratory (22 ± 1 °C).

## 3. Results and Discussion

### 3.1. Structure of MWCNTs

To investigate the crystal structure of MWCNTs, Raman spectroscopy was employed (Figure 4). A common way to quantify the amount of different functional groups in nanocarbon is to determine the I_D_/I_G_ ratio [77]. The former feature (D-band) informs about the presence of defects in the form of crystal dislocation or grafted functional groups, which generally correspond to the content of sp^3^ carbon atoms. The latter peak, on the other hand (G-band), originates from the vibration of neat sp^2^ carbon atoms in the lattice.

The results showed a typical amount of defects, as expected from MWCNTs of technical quality. The I_D_/I_G_ ratio was found to be relatively high and amounted to 1.68, which stayed in accordance with other results from the literature [78,79]. The NC7000^TM^ MWCNTs used for this study is a material produced in tonnes, which, despite its reduced crystallinity, makes it appropriate for large-scale applications, such as in composites for civil engineering, due to its abundance and reasonable price-to-quality ratio.

### 3.2. Dispersion of MWCNTs

The tests showed that the admixture type played an important role in the homogenization process. Surfactants can adsorb on the surface of cement grains and CNTs through electrostatic interactions, increasing the hydrophobicity of the grains and modifying the intermolecular attraction, thus modifying the rheological behavior and matrix hydration reaction [80,81]. In the case of the SP1-SP4 admixtures, regardless of the processing conditions (Table 1), proper dispersion could not be obtained in any of the cases. An increase in sonication time/power or the amount of admixture was insufficient to facilitate the individualization of MWCNTs. Upon the sonication completion, the nanocarbon content rapidly deposited on the bottom and on the walls of the containers (Figure 5b). These superplasticizers were primarily made of acrylic or (carboxylate ether) polymers, which did not have an appropriate affinity for MWCNTs to overcome their tendency to agglomerate as a result of van der Waals forces.

However, when the PL1 plasticizer based on lignosulfonates was employed, the results were very encouraging. In both of the explored cases (Table 4), the admixture facilitated the complete dispersion of MWCNTs in water (Figure 5). No MWCNT deposits could be discerned after the sonication was completed. Cellulose analogs are known for their excellent capabilities to solubilize MWCNTs [82,83,84], which could explain why these plasticizers effectively dispersed MWCNTs in this medium.

The admixture gives the cement particles a negative potential due to the effect of the separation of sulfo groups [85,86]. Surfactant molecules adsorbed on the MWCNT surface maintain colloid stability by electrostatic repulsion between the electric charges of their functional groups, and their dispersion efficiency depends on the length of the chains of the surfactant used [61]. It is important to stress that, according to the manufacturer, the addition of such a plasticizer results in lowering the surface tension, which can reduce the amount of required batched water to 12% or less. Another positive impact resulting from its addition to fresh concrete involves its plasticizing (as the name suggests) as well as optimizing the cement setting time delay due to the presence of a certain amount of saccharides in the formulation [87]. One also needs to keep in mind that lignosulfonates themselves can delay setting time by creating coatings on the surfaces of one of the hydration products—the C-S-H phase [88]. In light of the foregoing, the next step was to evaluate the properties of nanoconcrete containing both the MWCNTs and the specified plasticizer used for their dispersion.

### 3.3. Consistency and Setting Time

In the graphs shown in Figure 6 and Figure 7, the means of three consistency measurements for each of the mortar and fresh concrete samples are reported. In the former case, one can observe a decrease in mortar flow diameter with the increase in the amount of MWCNTs.

There is a 34 mm difference between the mortar that is free of MWCNTs (Z0, 0% with respect to cement), and the mortar rich in MWCNTs (Z4, 0.144% with respect to cement). The samples from Z0 to Z3 exhibited an average spread of over 200 mm; so, according to PN-EN 1015-6:2000 standards [89], they have the consistency of a liquid. On the other hand, the consistency of the Z4 sample according to the above standard can be classified as plastic.

The hydrophobic nature, the high specific surface area, and the strong van der Waals forces between individual MWCNTs negatively affect the workability of cement mixtures modified with nanomaterials [90]. The research shows that the workability of the mortar deteriorates with the increase in the amount of MWCNTs, and this trend follows earlier reports in [91,92,93]. However, the influence on the rheological properties presented herein is much smaller compared to the state of the art [94]. Therefore, the results obtained in this study confirm the suitability of PL1 plasticizer to disperse MWCNTs in the matrix.

At this point, it is important to describe the impact of surfactant-dispersed MWCNTs on the rheological behavior of cement paste [91,92,93], which, in turn, affects the workability of concrete [52]. Based on the research carried out on mortars, it can be concluded that, with the increase in the amount of nanomaterial used, the water demand for mixing increases, which may stem from the notable surface area of the MWCNTs, which may trap water and other species of the mixture. The results show that the difference in consistency is two classes of consistency with respect to the composition without MWCNTs (Z0–Z3—consistency of liquid and Z4—plastic consistency). Theoretically, it would be possible to use a lower w/c ratio when using a superplasticizer instead of a plasticizer to alleviate this issue. However, research on the dispersion of CNTs in water using an acrylate-based superplasticizer surfactant and a polycarboxylic ether did not yield positive results.

The characterization of fresh concrete presented in Figure 6 shows the negative effect of the addition of MWCNTs on the slump flow. According to the PN-EN 12350-5 standard, fresh concretes with a spread below 340 mm, as characterized by a B1 test (334 cm), have a consistency of F1. The sample B0 (468 cm) exhibited a consistency of F3, as determined for the flow range 420–480 mm. Based on the research carried out on mortars and fresh concretes, it can again be concluded that, with the increase in the amount of added MWCNTs, the water demand for mixing increased. Both in mortar (Z4) and concrete (B1), the same percentage of MWCTs caused the workability to deteriorate by approximately 15% and 28%, respectively.

Figure 8 shows the achieved setting times of the mortars depending on the amount of MWCNTs. The shortest setting time of 625 min was for the sample without MWCNTs (Z0). When MWCNTs were added, the setting time was longer, but no trend could be discerned as the content of MWCNTs was increased. The longest setting time was achieved for the Z1 sample with the amount of 0.036% MWCNT in relation to the cement weight and amounted to 677 min. The setting times of the other samples were at a comparable level, taking into account the uncertainty values. It was previously shown that the addition of CNTs [95] decreases the initial setting time of cement mortar since the presence of CNTs accelerates the rate of hydration of initial C_3_A (ettringite) by acting as a nucleating agent. Consequently, the stiffening and setting characteristics of cement paste primarily depend on hydration [95]. However, it is important to mention that the variation in setting time with the change in concentration of MWCNT between 0.1% and 0.3% was little. In the presented case, when a small amount of MWCNTs was added along with the plasticizer, the trend was not clear, which stays in accordance with the previously indicated findings. Regarding the observed delay in the setting time, the PL1 admixture with added MWCNTs may promote the formation of coatings on the surface of one of the hydration products (C-S-H phase) [88], which could explain the obtained results.

### 3.4. Mechanical Properties

The flexural strengths and compressive strengths of the mortars after 1, 7, and 28 days are shown in Table 7 and Figure 9, as well as Table 8 and Figure 10, respectively. In the first case, after 1 day, relatively low flexural strength results were obtained. This result could be justified by the nature of the admixture, which delayed the setting time. Overall, after 28 days, the use of 0.036% MWCNTs in relation to the cement mass (Z1) increased the flexural strength by about 4.5%. Previous research revealed that functionalized MWCNTs may reinforce the cement mortar when it is effectively dispersed in the matrix [96]. Furthermore, microscopic analysis showed that CNTs can potentially improve the quality of the cement matrix in the aggregate–paste interfacial transition zone, which improves the strength [97]. A considerable amount of work was devoted to elaborating the impact of CNTs on mechanical characteristics [38,40,41,98,99]. Li et al. investigated the mechanical behavior of cement pastes reinforced with 0.4–0.5% surface-modified CNTs and found a 19–25% improvement in bending and compressive strength [38]. On the other hand, the incorporation of pristine CNTs in the same study led to a reduction in compressive strength compared to a sample without nanomaterial, as proper stress transfer was not established. Other researchers also described the low efficiency of CNT in enhancing the mechanical strength of cement pastes [99,100] or mortars [37] due to the aforementioned reason. Nevertheless, some were able to achieve strength improvements of up to 35% when CNTs were well dispersed in the cement matrix [52,98,101]. For instance, the studies in [97] show that the incorporation of 0.05–0.1% of CNTs was able to increase the flexural strength up to 33%. Similarly, adding functionalized MWCNTs as a bridging factor increases the flexural strength of nanocomposites stored in limestone-saturated water and sulfate solution by about 40% [96]. To sum up, the beneficial effect of CNTs was only observed when proper integration with the matrix could be obtained. For that to happen, CNTs had to contain defects from the synthesis stage to be modified afterward.

In this study, the increase in strength was moderate. On the one hand, MWCNTs had a considerable number of functional groups, which enabled proper integration with the matrix. On the other hand, the storage conditions, which were supposed to imitate the natural conditions prevailing on the construction site during concrete care, did not favor reaching the highest possible strength. The lowest flexural strength was achieved by mortar (Z3), which contained 0.108% of MWCNTs in relation to the weight of cement. The difference was merely 1% compared to the reference mortar. Although no significant change can be noted at this point, these results demonstrate that the setting time of the material can be prolonged without negatively affecting the flexural strength of the mortars.

Next, the characterization of the compressive strengths of the mortars was carried out (Table 8, Figure 9) [83]. In the tests after 1 day, relatively low results of compressive strength were obtained again due to the delay in the setting time caused by the admixture incorporation. After 28 days, the use of 0.036% MWCNTs in relation to the cement mass (Z1) increased the compressive strength by less than 1%. When a larger amount of MWCNTs was added, the compressive strength of the material suffered. The lowest strength was achieved by mortar (Z3), which contained 0.108% carbon nanotubes in relation to the weight of cement. In this case, the difference was nearly 17% compared to the reference mortar.

The results of the compressive strength of concretes are presented in Figure 11. In the tests of the compressive strength of concrete after 7 days, very similar results for concrete with MWCNTs (44.73 MPa) and without MWCNTs (44.94 MPa) were obtained, taking into account the measurement uncertainties. However, after 28 days, the concrete with MWCNTs (B1) achieved a strength that was nearly 5% higher compared to the concrete without MWCNTs (B0), i.e., 53.37 MPa vs. 50.86 MPa. The presence of the same amount of nanomaterial in mortar (Z4) and concrete (B1) caused a deterioration of the compressive strength after 28 days by about 9% in the case of mortar and an improvement in the compressive strength of concrete by about 5%.

Mortar can be considered to be concrete without coarse aggregate, which means that, from the physical point of view, there is no difference in the structure of mortar and concrete [102,103,104]. Numerous studies have shown that mortar testing allows us to predict the rheological properties of fresh concrete, especially in the case of SCC (Self-Compacting Concrete), which is usually characterized by a high mortar content [102,103,104,105,106,107]. However, these studies focused on the rheological parameters of the materials in the steady state and not on determining the strength parameters, especially with the addition of MWCNTs, which act at the cement–grain boundary. The type of nanofiller (particle shape, particle size, specific surface, concentration of dispersed phase) contributes to the improvement of the operational properties of concretes; however, the final properties of a cement composite containing inorganic particles always depend on the degree of filler dispersion and interactions at the interface between the components.

### 3.5. Electrical Properties

Results of the impedance measurements obtained for specimens with and without MWCNTs are presented in Figure 12a and Figure 12b, respectively. In these figures, the frequency increases from right to left across the plots. The impedance response comprises two distinct areas: (i) a low-frequency area associated with the polarization at the interface between the electrodes and the specimen and (ii) a high-frequency arc with a center depressed from the real axis representing the bulk response of the material.

The characteristic point (local minima) of the connection of the two areas is frequently used to conclude the condition of a specimen [108,109].

The plots represent the mean characteristics obtained from five measurements of each specimen. Error bars correspond to the standard deviation values, which did not exceed 3%. Hence, the accuracy of the LCR meter that is at the level of 0.05% can be neglected in the analysis. The local minima at the beginning of the curves (presented as an inset in Figure 11b) shift towards lower resistance for specimens with MWCNTs, which is due to the introduction of conductive MWCNTs. A much smaller spread between different samples is also visible for specimens filled with nanocarbon. These results also appear more coherent since the standard deviation is around twice as small as for neat concrete. It is supposed that better repeatability and smaller spread may be related to polarization phenomena that occur to a lesser extent in the MWCNT composite. Consequently, tracking damage by electrical means becomes more sensitive when the nanoconcrete based on MWCNTs is under investigation.

## 4. Conclusions

For cement nanocomposites containing MWCNTs, the appropriate dispersion of nanofillers is the basic factor determining the final properties of the composite. The ability to disperse CNTs in water solutions with cement admixtures is closely related to the chemical structure of the admixture and its basic mechanism of action. The presented results prove that the plasticizer based on lignosulfonate disperses MWCNT in the suspension in the best way among other chemical bases of admixtures. This result was also visible in the subsequent tests of consistency, setting time, strength characteristics, and electrical conductivity of cement composites. It is also worth emphasizing that the tests were conducted comprehensively on mortars, concrete mixes, and concretes. The selected plasticizer also contributed to preventing significant deterioration of the consistency and inhibition of the acceleration of the hydration process by MWCNT.

Moreover, the use of MWCNTs in the amount of 0.036% of the cement mass increased the flexural and compressive strength of mortars. This is the optimal dose of MWCNTs in this case (w/c = 0.5, PL = 1.2% c.m.). However, when the content of MWCNTs was increased to 0.144% in relation to the cement mass, a decrease in the compressive strength of the mortar was observed. MWCNTs’ mechanical characteristics are predestined to enhance the flexural or tensile strength of a material rather than its compressive characteristics, as they tend to collapse under pressure [110].

In addition, the amount of MWCNT in Z4 (0.144% in relation to the cement mass, w/c = 0.5, PL = 1.2% c.m.) could be excessive, which could have an overall negative impact on the properties of the nanocomposite.

Nevertheless, the use of MWCNTs leads to a smaller spread of electrical impedance values. Greater repeatability in measuring the impedance of the composite positively influences the possibility of an electrical evaluation of the condition of the concrete. It is supposed that better repeatability and smaller spread may be related to polarization phenomena that occur to a lesser extent in MWCNT-filled nanoconcrete composite due to their electrical properties. In light of these findings, the concept developed in this study can be considered validated.

Due to the rich chemistry of carbon, there may be many more chemicals typically used for civil engineering that could disperse MWCNTs. Finding them would enable the incorporation of CNTs and other forms of nanocarbon into concrete while ensuring that its properties are not deteriorated due to the presence of unconventional additives, such as sodium dodecyl sulfate or sodium dodecylbenzene sulfonate, which are routinely used to make carbon nanostructures water compatible. Such an approach can considerably increase the technology-readiness level of smart concretes, thereby bringing them closer to wide commercial implementation [59,111].

## Figures and Tables

**Figure 1 materials-17-04972-f001:**
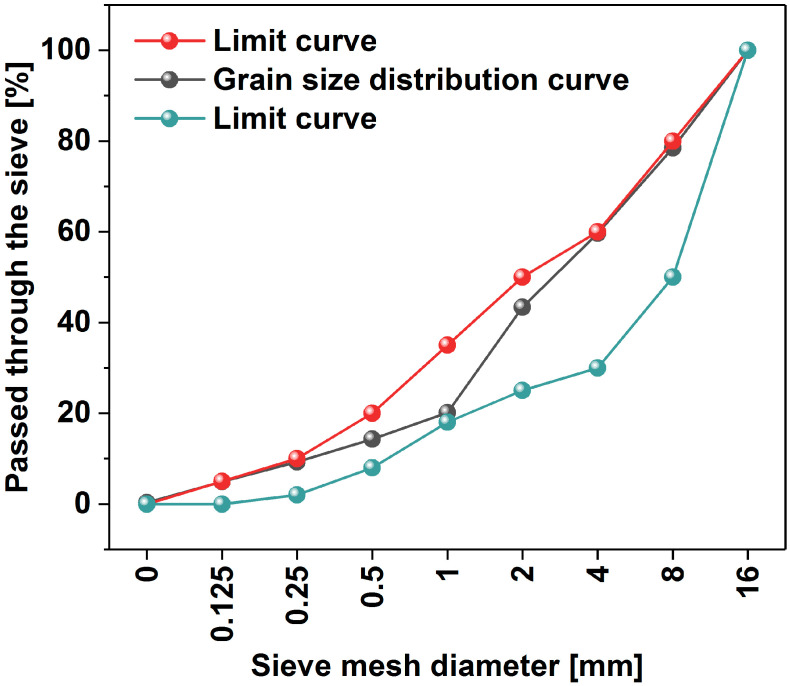
The aggregate particle size curve.

**Figure 2 materials-17-04972-f002:**
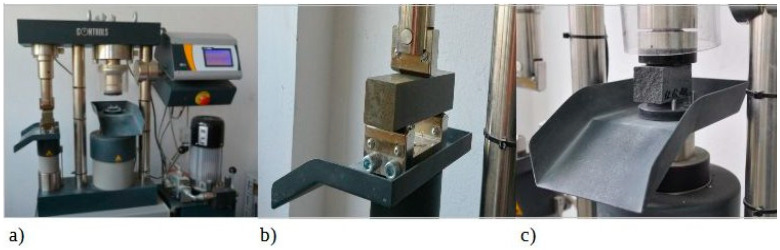
(**a**) Setup for carrying out strength tests (300/15 kN PILOT Automatic Compression-Flexural Cement Testers, model: 65-L1142, Controls company, Tucker, GA, USA); image of (**b**) flexural and (**c**) compressive strength of mortar tests.

**Figure 3 materials-17-04972-f003:**
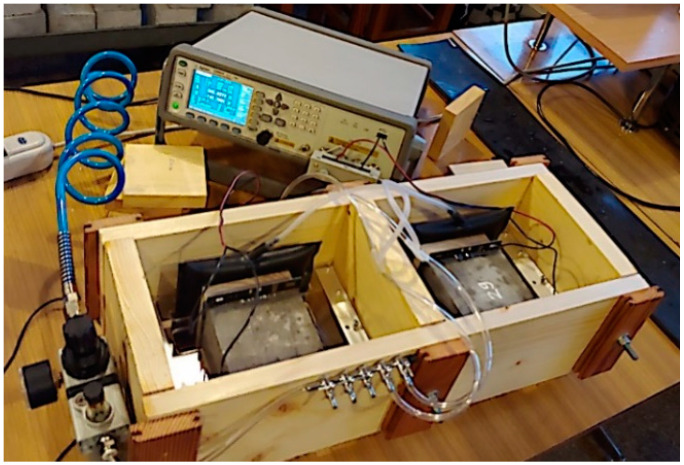
The setup used for electrical impedance measurements.

**Figure 4 materials-17-04972-f004:**
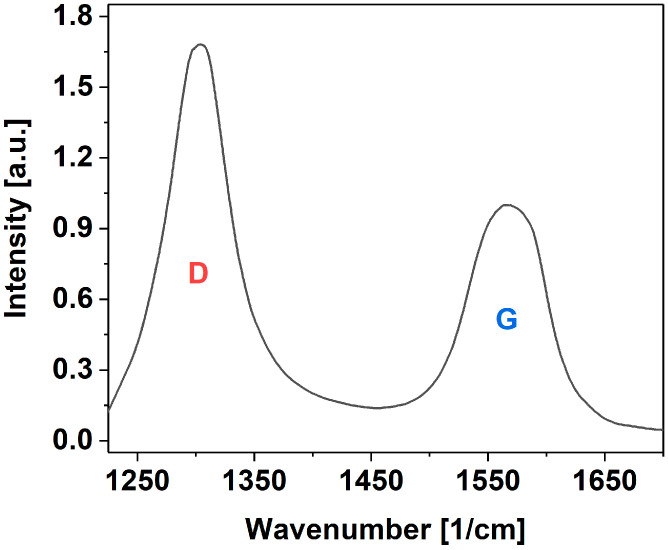
Raman spectrum of MWCNT powder used for this study (D—means D-band, G—means G-bands).

**Figure 5 materials-17-04972-f005:**
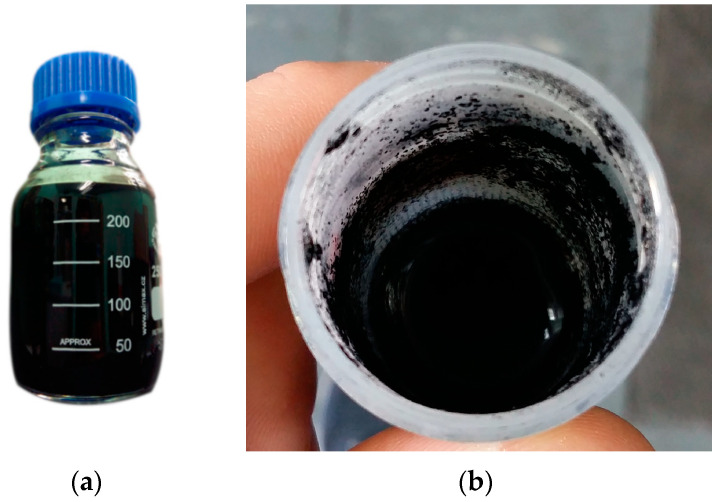
The MWCNTs dispersion in water facilitated by the concrete admixture: (**a**) PL1 (Sample #18 prepared according to the conditions reported in Table 4) and (**b**) SP3 (Sample #18).

**Figure 6 materials-17-04972-f006:**
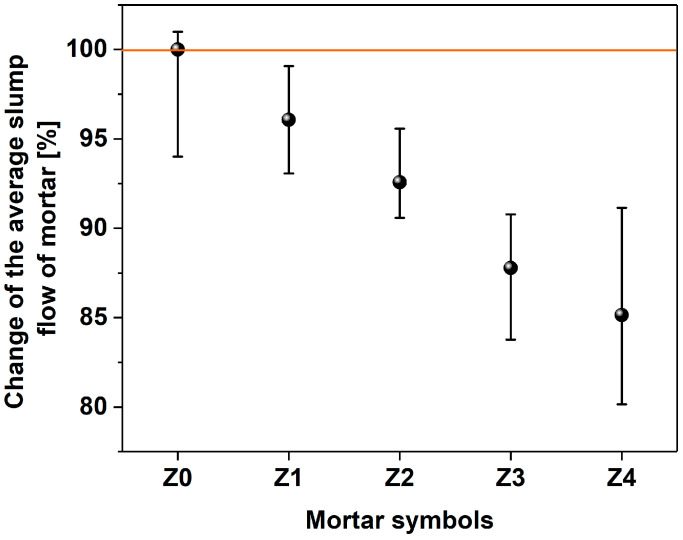
Slump flow of mortars without (Z0) and with MWCNTs (Z1–Z4, increasing content).

**Figure 7 materials-17-04972-f007:**
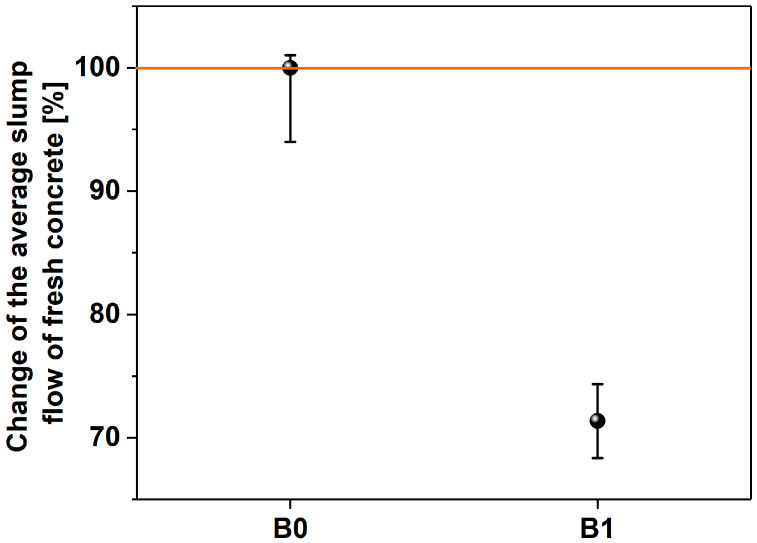
Slump flow of fresh concrete without (B0) and with the MWCNTs (B1).

**Figure 8 materials-17-04972-f008:**
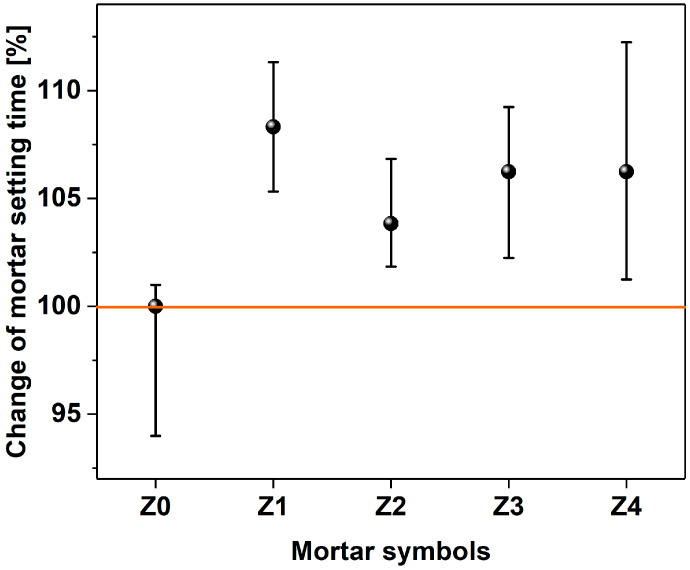
Setting time of mortars without (Z0) and with MWCNTs (Z1–Z4; increasing content).

**Figure 9 materials-17-04972-f009:**
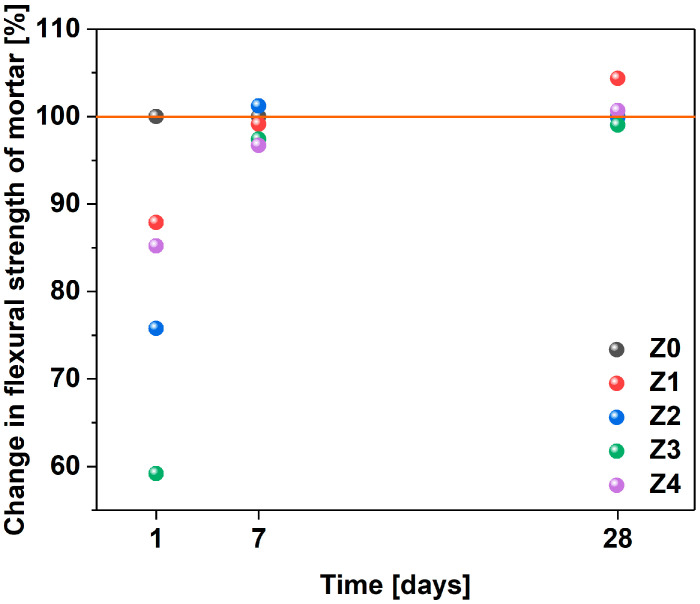
Specification of the flexural strength of mortars after 1, 7, and 28 days in relation to the sample free of MWCNTs (Z0). The Z1–Z4 samples contain MWCNTs and are arranged in ascending order in terms of amount.

**Figure 10 materials-17-04972-f010:**
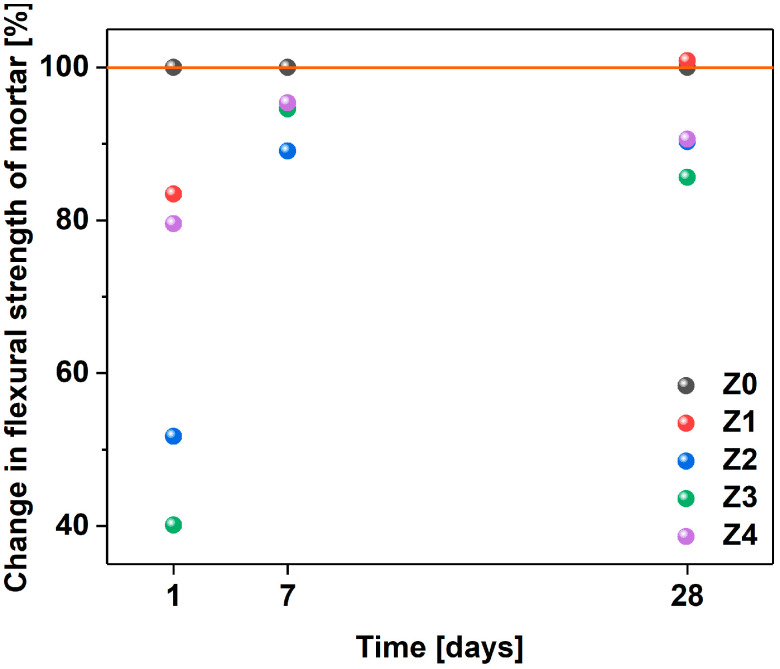
Comparison of compressive strength of mortars after 1, 7, and 28 days in relation to the sample free of MWCNTs (Z0). The Z1–Z4 samples contain MWCNTs and are arranged in ascending order in terms of amount.

**Figure 11 materials-17-04972-f011:**
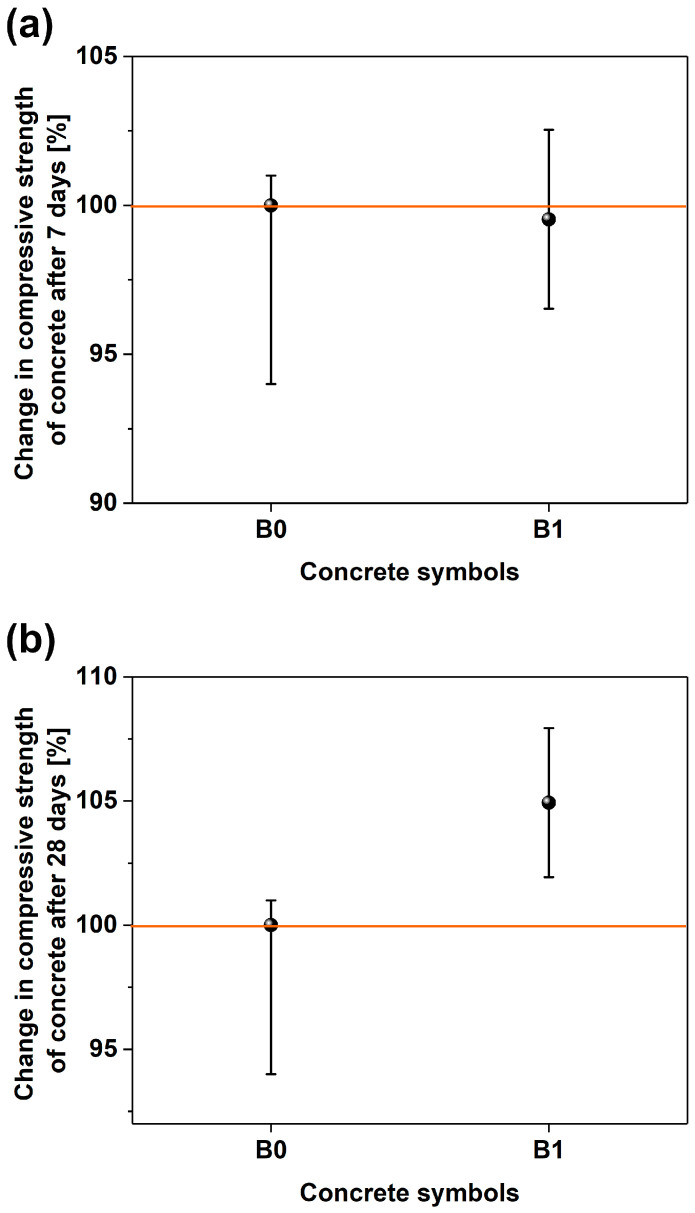
Graph of the compressive strength of concrete after (**a**) 7 days and (**b**) 28 days in the sample reference free of MWCNTs (B0).

**Figure 12 materials-17-04972-f012:**
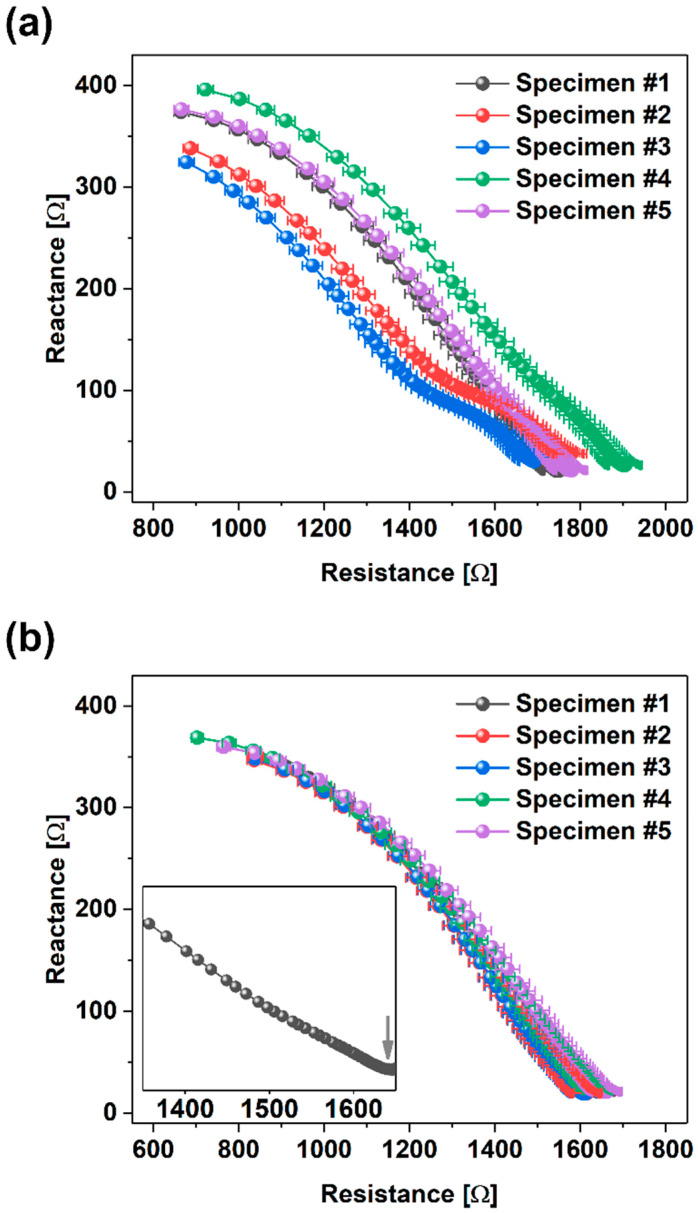
Nyquist plots of electrical impedance; (**a**) for specimens without MWCNTs (B0); (**b**) for specimens with MWCNTs (B1). The inset shows the electrical impedance of Specimen #1 with a characteristic local minimum indicated with an arrow.

**Table 1 materials-17-04972-t001:** Composition of CEM I 42.5 R cement (GÓRAŻDŻE CEMENT SA—Heidelberg Materials).

Composition [wt%]								Loss on Ignition [wt%]	Insoluble Parts [wt%]	The Specific Surface Area [m^2^/kg]
SiO_2_	CaO	Al_2_O_3_	Fe_2_O_3_	MgO	Na_2_O	SO_3_	K_2_O			
21.6	64.41	4.46	2.24	1.25	0.4	3.1	0.91	2.0	0.4	383

**Table 2 materials-17-04972-t002:** Physical and mechanical properties of the CEM I 42.5 R cement.

Properties		Results
Changes in volume, Le Chatelier		0.0 mm^3^
Beginning of setting time		175 min
Compressive strength	After 2 days	27.6 MPa
	After 28 days	53.4 MPa

**Table 3 materials-17-04972-t003:** Properties of superplasticizers and plasticizers.

Admixture	Chemical Base	Density [g/cm^3^]	pH at 20 °C	Chloride Content [wt%]	Alkali Content [wt%]
MAPEI (SP1)	acrylic polymers	1.06 ± 0.02	6.5 ± 1.0	≤0.1%	≤3.0%
MAPEI (SP2)	acrylic polymers (without formaldehyde)	1.06 ± 0.02	7.0 ± 1.0	≤0.1%	≤2.0%
BASF (SP3)	(carboxylate ether) polymers	1.06 ± 0.02	6.0 ± 1.0	≤0.1%	≤0.6%
BASF (SP4)	(carboxylate ether) polymers	1.07 ± 0.02	6.5 ± 1.0	≤0.1%	≤1.5%
MAPEI (PL1)	lignosulfonate	1.16	8.5	≤0.1%	≤6.0%

**Table 4 materials-17-04972-t004:** Details of the sonication process of the aqueous admixture mixture with MWCNTs. Successful formulations are shaded in grey.

No.	Symbol of Admixture	The Name of the Admixture	The Manufacturer	Amount of MWCNTs [g]	Amount of Admixture [g]	Amount of Water [g]	Sonication Power [%]	Sonication Time [min]
1	SP1	Dynamon NRG 1022	MAPEI (Gliwice, Poland)	0.250	0.250	25	100%	03:20
2	SP1	Dynamon NRG 1022	MAPEI	0.060	0.090	25	40%	03:20
3	SP1	Dynamon NRG 1022	MAPEI	0.125	0.250	25	40%	03:20
4	SP1	Dynamon NRG 1022	MAPEI	0.125	0.500	25	40%	03:20
5	SP1	Dynamon NRG 1022	MAPEI	0.125	1.000	25	40%	03:20
6	SP1	Dynamon NRG 1022	MAPEI	0.060	0.125	25	40%	03:20
7	SP1	Dynamon NRG 1022	MAPEI	0.125	0.125	25	40%	04:20
8	SP1	Dynamon NRG 1022	MAPEI	0.030	0.040	25	40%	30:00
9	SP2	Dynamon SR1	MAPEI	0.125	0.125	25	40%	03:20
10	SP2	Dynamon SR1	MAPEI	0.125	0.250	25	40%	03:20
11	SP2	Dynamon SR1	MAPEI	0.125	0.500	25	40%	03:20
12	SP2	Dynamon SR1	MAPEI	0.125	1.000	25	40%	03:20
13	SP2	Dynamon SR1	MAPEI	0.060	0.060	25	40%	03:20
14	SP2	Dynamon SR1	MAPEI	0.060	0.125	25	40%	03:20
15	SP2	Dynamon SR1	MAPEI	0.600	1.250	250	60%	120:00
16	SP3	MasterGlenium ACE 430	BASF (Warsaw, Poland)	0.030	0.030	25	40%	30:00
17	SP4	MasterGlenium SKY 591	BASF	0.030	0.040	25	40%	30:00
18	PL1	Mapeplast BV 34	MAPEI	0.030	0.050	25	40%	30:00
19	PL1	Mapeplast BV 34	MAPEI	0.600	1.250	250	60%	80:00

**Table 5 materials-17-04972-t005:** Composition of mortars (% c.m.—percent of cement mass).

Symbol of Mortar	Z0	Z1	Z2	Z3	Z4
MWCNTs [% c.m.]	0	0.036	0.072	0.108	0.144
MWCNTs [g]	0	0.162	0.324	0.486	0.648
w/c ratio	0.5				
Cement [g]	450				
PL1 [% c.m.]	1.2				

**Table 6 materials-17-04972-t006:** Composition of concrete for 1 m^3^.

Symbol of Concrete	B0	B1
MWCNTs [% c.m.]	0	0.144
MWCNTs [kg]	0	0.504
w/c ratio	0.5	0.5
Cement [kg]	350	350
Sand [kg]	500	500
Aggregate 2–8 mm [kg]	670	670
Aggregate 8–16 mm [kg]	625	625
PL1 [% c.m.]	1.2	1.2

**Table 7 materials-17-04972-t007:** The results of mortars flexural strength.

Symbol of Mortar	Flexural Strength of Mortars		
	After 1 Day [MPa]	After 7 Days [MPa]	After 28 Days [MPa]
Z0	0.74 ± 0.031	6.57 ± 0.24	7.47 ± 0.34
Z1	0.65 ± 0.021	6.52 ± 0.21	7.80 ± 0.35
Z2	0.56 ± 0.027	6.65 ± 0.29	7.47 ± 0.33
Z3	0.44 ± 0.021	6.40 ± 0.27	7.40 ± 0.32
Z4	0.63 ± 0.030	6.36 ± 0.25	7.52 ± 0.35

**Table 8 materials-17-04972-t008:** The results of mortar compressive strength.

Symbol of Mortar	Compressive Strength of Mortars					
	After 1 Day [MPa]	Average Strength after 1 Day [MPa]	After 7 Days [MPa]	Average Strength after 7 Days [MPa]	After 28 Days [MPa]	Average Strength after 28 Days [MPa]
Z0	3.23	3.06 ± 0.45	43.70	42.40 ± 1.89	39.80 *	55.06 ± 2.58
	3.33		42.14		57.49	
	3.15		44.78		51.23	
	3.63		43.33		53.67	
	2.49		40.17		56.86	
	2.54		40.27		56.04	
Z1	3.18	2.56 ± 0.53	36.16	40.42 ± 2.56	42.11 *	55.54 ± 3.09
	2.90		40.46		57.29	
	1.89		44.10		56.22	
	1.93		41.43		59.31	
	2.63		40.08		51.63	
	2.80		40.27		53.27	
Z2	1.33	1.58 ± 0.23	36.77	38.02 ± 1.90	51.45	49.72 ± 2.01
	1.34		39.47		49.82	
	1.74		40.40		51.13	
	1.9		35.45		46.46	
	1.67		39.07		51.25	
	1.52		36.98		48.20	
Z3	1.76	1.28 ± 0.58	38.83	40.39 ± 3.21	44.23	47.13 ± 4.52
	1.67		38.61		45.73	
	0.51		37.98		44.90	
	1.58		37.89		42.50	
	0.56		44.84		51.38	
	1.58		44.16		54.02	
Z4	2.87	2.44 ± 0.84	43.79	40.72 ± 3.38	53.26	49.88 ± 2.74
	2.95		42.24		52.26	
	1.10		38.37		50.75	
	1.66		35.93		47.51	
	3.05		44.60		49.29	
	2.98		39.40		46.18	

* Rejected result—result is not included in the average because of the large deviation from the mean.

## Data Availability

The original contributions presented in the study are included in the article, further inquiries can be directed to the corresponding author.

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
