# Peer review of "Facile Incorporation of Carbon Nanotubes into the Concrete Matrix Using Lignosulfonate Surfactants"

_materials, 2024, doi:10.3390/ma17204972_

Round 1

Reviewer 1 Report

Comments and Suggestions for Authors

The paper is very interesting and includes new and prospective results. The paper can be accepted for publication after major revisions; the following issues should be clarified. The compatibility of the nanotubes with additives and matrixes (polymer or concrete) is the main factor in the successful fabrication of the various composites.  

Please add more detailed information to the admixture specification. What is their nomenclature in the BASF or Mapei? Please add the appropriate specifications.

It will be valuable to provide the XRD analysis for MWCNT dispersions to define the d-spaces for dispersed nanotubes. 

It is really hard to understand why lignosulfonates are compatible with carbon nanotubes. At least an appropriate discussion should be provided. I suggest using the relevant paper, https://doi.org/10.1016/j.apsadv.2021.100104 where similar issues were discussed. Why were MWCNTs incompatible with other types of admixtures? 

Comments on the Quality of English Language

Minor editing of English language required.

Author Response

The paper is very interesting and includes new and prospective results. The paper can be accepted for publication after major revisions; the following issues should be clarified. The compatibility of the nanotubes with additives and matrixes (polymer or concrete) is the main factor in the successful fabrication of the various composites.  

Thank you for your sincere review. We addressed all your comments carefully. We have great hope to reconsider our changes.

  1. Please add more detailed information to the admixture specification. What is their nomenclature in the BASF or Mapei? Please add the appropriate specifications – the manufacturer's name and trade name of the products have been added, all information has now become public and can now be easily found in online sources. - In subchapter 2.1 in Table 1, Table 3 and Table 4 information about the cement manufacturer and chemical admixtures and trade names has been added. The cement manufacturer is GÓRAÅ»DÅ»E CEMENT SA -Heidelberg Materials and the admixtures come from MAPEI (Dynamon NRG 1022, Dynamon SR1, Mapeplast BV 34) and BASF (MasterGlenium ACE 430,MasterGlenium SKY).

  2. It will be valuable to provide the XRD analysis for MWCNT dispersions to define the d-spaces for dispersed nanotubes. – we agree with the Reviewer that XRD gives insightful information about the crystallographic structure of MWCNTs. Unfortunately, at the moment, we do not have access to characterize the material this way. Nonetheless, based on the characterization conducted in this manuscript, we do not expect any major changes to the state of the dispersed CNTs in the concrete matrix. According to our experience, the employed sonication conditions are mild and do not affect the nanocarbon material.

  3. It is really hard to understand why lignosulfonates are compatible with carbon nanotubes. At least an appropriate discussion should be provided. I suggest using the relevant paper, https://doi.org/10.1016/j.apsadv.2021.100104 where similar issues were discussed. Why were MWCNTs incompatible with other types of admixtures? – in the manuscript, we previously stated the following “However, when the PL1 plasticizer based on lignosulfonates was employed, the results were very encouraging. In both of the explored cases (Table 4), the admixture facilitated complete dispersion of MWCNTs in water (Fig. 5). No MWCNT deposits could be discerned after the sonication was completed. Cellulose analogues are known for their excellent capabilities to solubilize MWCNTs [75][76], which could explain why this plasticizers effectively dispersed MWCNTs in this medium.” Hence, it is clear that the cellulose analogues show some affinity to MWCNTs effectively coating them in the liquid medium through MWCNT individualization, enabling their suspension after overcoming the van der Waals interactions between MWCNTs. While revising this work, we found another reference which proves that lignosulfonates are good dispersants of CNTs, so we added it in the text along with the suggested reference. As also highlighted in the manuscript, are much more appropriate in this context since they are already used as concrete superplasticizers. In contrast, classical surfactants such as sodium dodecyl sulfate or sodium dodecylbenzene sulfonate may affect the properties of the concrete since they are not commonly used in this area.

Reviewer 2 Report

Comments and Suggestions for Authors

Dear authors,

The topic approached in your manuscript could be interesting from the point of view of the incorporation of carbon nanotubes into the concrete matrix using lignosulfonate surfactants in order to transform it in a smart concrete used as functional materials in different applications for which the electrical properties are also important like the mechanical properties.

In order to improve this manuscript, I have some questions and I would also like to recommend some major changes and improvement as it is shown below.

1. In sub-chapter 2.1, the authors should write the manufacturer and location of company for all materials used (CEM I 42.5 R cement, superplasticizers and plasticizer).

2. The authors must cite the references for all properties given in Table 1, Table 2 and Table 3 (the references should be the technical sheets of those materials). Otherwise, the authors should give the procedure use for testing.

3. At lines 127-128, it is not clear if the properties reported are corelated with the measuring units. The reference used should be also cited.

4. The chapters 2.2.8 and 2.2.9 should be sub-chapters of 2.2.7.

5. For the text located at lines 225-226, there is not any loading method or shape of the bent bars depicted below as is written in text. Anyway, the number of the figure referred should be clarified. Please, give more details about the shape and dimensions of the flexural specimens made of mortars. More details about the loading scheme are also required and also, the loading rate.

6. The authors should clarify why the compressive tests were made on the halves of the bars obtained from the flexural tests. It is not clear. Are the compressive specimens obtained after flexural tests? But the flexural tests affects the specimen material and it cannot be reloaded. Please, make clarification. In text (lines 229-230) it is also written that the specimens and method is shown below.

7. What is the loading rate (speed) in compression tests applied to the mortar specimens?

8. How is the force applied on the mortar and concrete specimens in the compressive tests? How much is the area where the distributed force is applied?

9. The authors should give more details about the testing machine used in the compressive test and in the flexural test: type, manufacturer, some technical characteristics.

10. The authors should give the stdev for data reported in Table 7 and Table 8.

11. Regarding the mechanical properties, the authors should write at least one conclusion about the effects of MWCNTs content on the mechanical properties of the concrete types investigated in Conclusion section.

12. In Conclusion section, the authors should recommend the best proportion for the constituents used in mortar and concrete in order to obtain the best balance between electrical and mechanical properties.

Comments on the Quality of English Language

Dear authors,

In my opinion, the text is quite well written in English, only small corrections are needed.

Author Response

Dear authors,

The topic approached in your manuscript could be interesting from the point of view of the incorporation of carbon nanotubes into the concrete matrix using lignosulfonate surfactants in order to transform it in a smart concrete used as functional materials in different applications for which the electrical properties are also important like the mechanical properties. In order to improve this manuscript, I have some questions and I would also like to recommend some major changes and improvement as it is shown below.

Thank you for your sincere review. We addressed all your comments carefully. We have great hope to reconsider our changes.

  1. In sub-chapter 2.1, the authors should write the manufacturer and location of company for all materials used (CEM I 42.5 R cement, superplasticizers and plasticizer) - In subchapter 2.1 in Table 1, Table 3 and Table 4 information about the cement manufacturer and chemical admixtures and trade names has been added. The cement manufacturer is GÓRAÅ»DÅ»E CEMENT SA -Heidelberg Materials and the admixtures come from MAPEI (Dynamon NRG 1022, Dynamon SR1, Mapeplast BV 34) and BASF (MasterGlenium ACE 430,MasterGlenium SKY).

  1. The authors must cite the references for all properties given in Table 1, Table 2 and Table 3 (the references should be the technical sheets of those materials). Otherwise, the authors should give the procedure use for testing – the manufacturer's name and trade name of the products have been added, all information has now become public and can now be easily found in online sources.

  1. At lines 127-128, it is not clear if the properties reported are corelated with the measuring units. The reference used should be also cited – Our research results were mainly influenced by the chemical basis of the admixture, however, it can also be seen that admixtures with the same chemical basis but different pH or alkali content have no effect on the degree of dispersion.

  1. The chapters 2.2.8 and 2.2.9 should be sub-chapters of 2.2.7. – corrected.

  1. For the text located at lines 225-226, there is not any loading method or shape of the bent bars depicted below as is written in text. Anyway, the number of the figure referred should be clarified. Please, give more details about the shape and dimensions of the flexural specimens made of mortars. More details about the loading scheme are also required and also, the loading rate – corrected. Detailed answer is given below.

  1. The authors should clarify why the compressive tests were made on the halves of the bars obtained from the flexural tests. It is not clear. Are the compressive specimens obtained after flexural tests? But the flexural tests affects the specimen material and it cannot be reloaded. Please, make clarification. In text (lines 229-230) it is also written that the specimens and method is shown below – corrected. Detailed answer is given below.

  1. What is the loading rate (speed) in compression tests applied to the mortar specimens? –corrected. Detailed answer is given below.

  1. How is the force applied on the mortar and concrete specimens in the compressive tests? How much is the area where the distributed force is applied? – corrected. Detailed answer is given below.

  2. The authors should give more details about the testing machine used in the compressive test and in the flexural test: type, manufacturer, some technical characteristics. – corrected. Detailed answer is given below.

The following information was included in the manuscript in response to remarks 5-9.

2.2.7. The methodology of strength evaluation

The strength test involving flexural and compression of the hardened mortar was carried out in line with the standard PN-EN 1015-11:2001 after 28 days, and additionally after 1 and 7 days (Fig.2). The strength test involving compression of the hardened concrete was carried out in line with the standard PN-EN 12390-1:2013-03 after 7 and 28 days.

c)

b)

a)

Figure 2. Device for carrying out strength tests: a) All equipment from CONTROLS;
b) Flexural strength of mortar test; c) compressive strength of mortar test.

2.2.7.1 Evaluation of the flexural strength of mortar

To determine the flexural strength, the bars are placed on cylindrical supports with a diameter of 10 mm and a span of 100 mm and the sample is loaded with a centrally located cylinder of the same length and diameter as the supports. The specimen undergoing flexural strength testing is shown in Figure 2 b. The bars were loaded to the point of breaking. The concentrated force acted in the middle of the span.

2.2.7.2 Evaluation of the compressive strength of mortar and concrete

The compressive strength test of the mortar was carried out on the halves of the bars obtained from the flexural tests.

Concrete samples were formed in accordance with the PN-EN 12390-1:2013-03 standard. Three samples were made for each test date. The molded samples in plastic molds. Samples with dimensions of 10 x 10 x 10 cm were made. The samples were disassembled after 1 day and were stored in a climate chamber with water at a temperature of 20±2oC until testing. The strength test was carried out using the testing machine (Fig. 2c). The load was applied perpendicular to the forming direction. The speed of the applied load was 0.6±0.2 MPa/s. The destructive force is transmitted using metal washers measuring 40 mm x 40 mm and 10 mm thick. The top plate is fitted with holes for guide pins. It is supported by springs. In the center of each plate there is a rectangular plate. They are placed in such a way that when the upper plate is pressed against the lower plate, the surfaces overlap. The halves of the bars, which are obtained after bending, are placed symmetrically on the surface of the plate. The pressure increase per sample should be 15-20 kg/cm2 per second. When crushing the sample, it is important to read the pressure on the pressure gauge.
The measure of compressive strength is nothing else than the ratio of the force indicated by the dynamometer or corresponding to the greatest pressure indicated by the manometer to the surface area. The compressive strength of the mortar is the arithmetic average of the strength of the six halves. If one of the results is 5% greater than the calculated average, the result is incorrect. It should be discarded and the average of the remaining results calculated. The test should be repeated when two results differ from the calculated mean by more than 5%.

  1. The authors should give the stdev for data reported in Table 7 and Table 8. – standard deviation values were added to Table 8. In case of Table 7, only one measurement for every sample was made, so it is not possible to determine the standard deviation values.

  1. Regarding the mechanical properties, the authors should write at least one conclusion about the effects of MWCNTs content on the mechanical properties of the concrete types investigated in Conclusion section Thank you for your comment. The Conclusions section has been modified and a conclusion has also appeared regarding the strength of cement composites with nanomaterial used in the study. Moreover, the use of MWCNTs in the amount of 0.036% of the cement mass increased the flexural and compressive strength of mortars. This is the optimal dose MWCNTs in this case (w/c=0,5, PL=1.2% c.m.). However, when the content of MWCNTs was increased to 0.144% of in relation to the cement mass, a decrease in the compressive strength of the mortar was observed. MWCNTs’ mechanical characteristics are predestined to enhance the flexural or tensile strength of a material, rather than its compressive characteristics, as they tend to collapse under pressure [107].

What is more, the amount of MWCNTs in Z4 (0.144% of in relation to the cement mass, w/c=0,5, PL=1.2% c.m.), could be excessive, which could make an overall negative impact on the properties of the nanocomposite.

  1. In Conclusion section, the authors should recommend the best proportion for the constituents used in mortar and concrete in order to obtain the best balance between electrical and mechanical properties Thank you for your comment. Best proportion for the constituents used in cement composite was given in the conclusions section as above.

Reviewer 3 Report

Comments and Suggestions for Authors

General Comments

The authors carried out work on an interesting subject matter.

However, the authors do not appear to present sufficient proof to support their important claims and exclude other contributary factors to their observations.

There is no proof from characterization of mixed materials that confirm dispersal of the MWCNTs dispersed in water (Figure 4) is maintained on mixing with concrete.

There are no images (micro and/or micro) of the different concretes and mortars to permit readers” appreciation of the possible effects of CNT incorporation on the presentation (e.g. porosity) of the different admixture  both before and after mechanical testing. Authors should consult the following to see how characterization data are to be presented:

(a)  Figure 6 of Zhang et al (2020) to see how such can be presented:  Zhang, J., Ke, Y., Zhang, J., Han, Q., & Dong, B. (2020). Cement paste with well-dispersed multi-walled carbon nanotubes: Mechanism and performance. Construction and Building Materials, 262, 120746. https://doi.org/10.1016/j.conbuildmat.2020.120746

(b)  Evangelista, A. C. J., de Morais, J. F., Tam, V., Soomro, M., Di Gregorio, L. T., & Haddad, A. N. (2019). Evaluation of carbon nanotube incorporation in cementitious composite materials. Materials, 12(9), 1504. https://doi.org/10.3390/ma12091504

(c)   Páez-Pavón, A., García-Junceda, A., Galán-Salazar, A., Merodio-Perea, R. G., Sánchez del Río, J., & Lado-Touriño, I. (2022). Microstructure and electrical conductivity of cement paste reinforced with different types of carbon nanotubes. Materials, 15(22), 7976. https://doi.org/10.3390/ma15227976

Complementary methods can be used to evaluate the electrical properties. representative "native" plots from mechanical tests need to be provided as supplementary materials

Some sections of the manuscript need to be re-phrased to improve clarity.

Suggestions to remedy these and other errors have been provided  to the authors  under the specific comments below.

Specific Comments

LINE 11: Write “One of the ways to turn concrete into  smart concrete” instead of “One of the ways how a concrete can be turned into a smart concrete”.

LINE 14-15: Write “often negatively impacts the characteristics of concrete” instead of “often makes a negative impact on the characteristics of concrete”.

LINE 16: Write “by using“ instead of “thanks to the application  of”

LINE 45: Write “are known to seal” instead of “seal”.

LINE 45: Delete “they”.

LINE 46: Delete “they”.

LINE 52: Delete “sources”.

LINE 82: Write “using” instead of “, when”

LINE 83: Write “and” instead of “are employed at“.

LINE 101-103: Rephrase this sentence as it is lacking in clarity. Avoid using “Because” to start a sentence.

LINE 107: Write “The broad spectrum of experiments conducted“ instead of “A broad spectrum of conducted experiments“.

LINE 115: In Table 2, “changes in Volume” do not appear to be in correct units; mm is for length and mm3 for volume.

LINE XXX: Xxxxxxxxxxx

LINE 147: Write “dispersion” instead of “dispersion”.

LINE 150: Please, in Table 4, can you clarify what is meant by “sonification amplitude” which was expressed in percentages.?

LINE 137-151: Authors should have provided rheological information of the different admixtures employed as this could be a credible contributor to the trends observed.

LINE 212: Write “concrete” instead of “a concrete”.

LINE 228-229: “The compressive strength test of the mortar was carried out on the halves of the bars obtained from the flexural tests.”

Why were halves from flexural tests used?

Would this testing procedure not affect the compressive test result? This should have been caried out on pristine samples.

Please  introduce sketches/illustrations of your test methods and test sequences to help the reader follow your work more easily.

LINE 269-271: Can you explain in the manuscript why it is suitable as claimed even with lower crystallinity?

LINE 274-275: “Surfactants can adsorb on the surface of cement grains and CNTs through electrostatic interactions, increasing the hydrophobicity of the grains.”

This is under results. The statement above is not supported by any experimental proof in this work. Appears like a conjecture.

LINE 278: “…..in none of the cases proper dispersion could be obtained”

What is the proof that proper dispersion was not obtained?

LINE 303:Write “mind” not “ming”.

LINE 309: Delete “consistency”.

LINE 332-333: "The results show that the difference inconsistency is two levels with respect to the composition without MWCNTs."

This sentence lacks clarity.

What is the exact meaning of the phrase “difference inconsistency is two levels”?

LINE 337: Write “yield” instead of “bring”.

LINE 343: Delete “the”.

LINE 350: Write “The shortest time of is 625 minutes was for the sample without MWCNTs (Z0).” Instead of “The shortest time was for the sample without MWCNTs (Z0) is 625 minutes”.

LINE 357: “C3A” Abbreviation is undefined.

LINE 457-458: The characteristic point (local minima) of the connection of the two areas is frequently used to conclude about condition of a specimen.

Citations are much needed to support this statement.

LINE 490-492: “That is because, MWCNTs’ mechanical characteristics are predestined to enhance the flexural or tensile rather than compressive properties.”

Predestined? How?

Comments on the Quality of English Language

Significant English editing is necessary.

Author Response

The authors carried out work on an interesting subject matter. However, the authors do not appear to present sufficient proof to support their important claims and exclude other contributary factors to their observations.

Thank you for your sincere review. We addressed all your comments carefully. We have great hope to reconsider our changes.

  1. There is no proof from characterization of mixed materials that confirm dispersal of the MWCNTs dispersed in water (Figure 4) is maintained on mixing with concrete – we agree with the Referee that obtaining a proper dispersion of CNTs in the concrete matrix is essential. That is why, we considered characterization of the composite material using SEM, but this would only show us the distribution of the CNTs on the surface. Concomitantly, the material is electrically conducting, and the value of electrical conductivity is quite appreciable, meaning that the CNTs in the material must have formed good percolation pathways. Based on this, with high likelihood, the dispersion of CNTs in the concrete matrix is high or, at least high enough for the concrete to have sensing capabilities.

  2. There are no images (micro and/or micro) of the different concretes and mortars to permit readers” appreciation of the possible effects of CNT incorporation on the presentation (e.g. porosity) of the different admixture both before and after mechanical testing. Authors should consult the following to see how characterization data are to be presented:

(a)  Figure 6 of Zhang et al (2020) to see how such can be presented:  Zhang, J., Ke, Y., Zhang, J., Han, Q., & Dong, B. (2020). Cement paste with well-dispersed multi-walled carbon nanotubes: Mechanism and performance. Construction and Building Materials, 262, 120746. https://doi.org/10.1016/j.conbuildmat.2020.120746

(b)  Evangelista, A. C. J., de Morais, J. F., Tam, V., Soomro, M., Di Gregorio, L. T., & Haddad, A. N. (2019). Evaluation of carbon nanotube incorporation in cementitious composite materials. Materials, 12(9), 1504. https://doi.org/10.3390/ma12091504

(c)   Páez-Pavón, A., García-Junceda, A., Galán-Salazar, A., Merodio-Perea, R. G., Sánchez del Río, J., & Lado-Touriño, I. (2022). Microstructure and electrical conductivity of cement paste reinforced with different types of carbon nanotubes. Materials, 15(22), 7976. https://doi.org/10.3390/ma15227976

We do agree with the Reviewer that observing the microstructure of the material provides additional details. Unfortunately, during the revision of the manuscript, we did not have the capacity to execute such measurements. Nonetheless, given the importance of this aspect, we believe that this topic deserves a separate investigation to study these effects in detail. We intend to do it in the near future. In the meantime, we cited these valuable references to inform the readers about the impact of such fillers on the material structure.

  1. Complementary methods can be used to evaluate the electrical properties. Representative "native" plots from mechanical tests need to be provided as supplementary materials. – it is true that the electrical properties of insulating materials (including concrete) can be tested using various methods. However, in this work, our goal was to use a method simply informing about the condition of the material, especially the formation of possible microcracks. One of the most common and convenient methods that work well in this type of research is impedance spectroscopy. The use of a wide frequency spectrum allows conclusions about the material based on Nyquist characteristics. Because of the availability of RLC bridges, it allows one to implement this method at relatively low costs, hence we only explored this option in this article, considering it as the most commercially relevant. We do not have any other results from the testing machine. The testing machine from CONTROLS gives the result immediately and for 6 samples of the same composition we calculate the average.

  2. Some sections of the manuscript need to be re-phrased to improve clarity.|- Some sections of the article have been significantly changed and improved, or supplemented. Chapter 2.2. Methods has been changed - the origin of materials and the description of the tests have been supplemented. The conclusions chapter has also been changed. The conclusions regarding the strength tests have been supplemented and information on the optimal composition of the cement composite with nanomaterial has been added.

  3. Suggestions to remedy these and other errors have been provided to the authors under the specific comments below.

LINE 11: Write “One of the ways to turn concrete into smart concrete” instead of “One of the ways how a concrete can be turned into a smart concrete” – corrected.

LINE 14-15: Write “often negatively impacts the characteristics of concrete” instead of “often makes a negative impact on the characteristics of concrete” – corrected.

LINE 16: Write “by using“ instead of “thanks to the application  of” – corrected.

LINE 45: Write “are known to seal” instead of “seal” – corrected.

LINE 45: Delete “they” – corrected.

LINE 46: Delete “they” – corrected.

LINE 52: Delete “sources” – corrected.

LINE 82: Write “using” instead of “, when” – corrected.

LINE 83: Write “and” instead of “are employed at“– corrected.

LINE 101-103: Rephrase this sentence as it is lacking in clarity. Avoid using “Because” to start a sentence – corrected.

LINE 107: Write “The broad spectrum of experiments conducted“ instead of “A broad spectrum of conducted experiments“– corrected.

LINE 115: In Table 2, “changes in Volume” do not appear to be in correct units; mm is for length and mm3 for volume– corrected.

LINE 147: Write “dispersion” instead of “dispersion” – corrected.

LINE 150: Please, in Table 4, can you clarify what is meant by “sonification amplitude” which was expressed in percentages? – this unfortunate expression was clarified by changing “amplitude” into “power” (power with respect to the maximum power this sonicator can deliver, i.e., 200W).

LINE 137-151: Authors should have provided rheological information of the different admixtures employed as this could be a credible contributor to the trends observed.

Thank you for your valuable attention. Our research consisted of several-stage verification aimed at selecting the appropriate admixture and optimal composition of the cement composite. In the first stage, the type of admixture and its chemical base were verified in order to select an admixture compatible with the tested MWCNTs. A photo showing the difference between admixtures in water suspension has also been added to the article. Apart from the lignosulfonate-based admixture, no other admixture satisfactorily formed a homogeneous dispersion with MWCNTs, which disqualified them from further rheological tests.

LINE 212: Write “concrete” instead of “a concrete”- corrected

LINE 228-229: “The compressive strength test of the mortar was carried out on the halves of the bars obtained from the flexural tests.”

Why were halves from flexural tests used?

Would this testing procedure not affect the compressive test result? This should have been caried out on pristine samples.

Please  introduce sketches/illustrations of your test methods and test sequences to help the reader follow your work more easily. – the following explanations and photos given below were added in text.

2.2.7. The methodology of strength evaluation

The strength test involving flexural and compression of the hardened mortar was carried out in line with the standard PN-EN 1015-11:2001 after 28 days, and additionally after 1 and 7 days (Fig.2). The strength test involving compression of the hardened concrete was carried out in line with the standard PN-EN 12390-1:2013-03 after 7 and 28 days.

a)

b)

c)

Figure 2. Device for carrying out strength tests: a) All equipment from CONTROLS; b) Flexural strength of mortar test; c) compressive strength of mortar test.

2.2.7.1 Evaluation of the flexural strength of mortar

To determine the flexural strength, the bars are placed on cylindrical supports with a diameter of 10 mm and a span of 100 mm and the sample is loaded with a centrally located cylinder of the same length and diameter as the supports. The specimen undergoing flexural strength testing is shown in Figure 2 b. The bars were loaded to the point of breaking. The concentrated force acted in the middle of the span.

2.2.7.2 Evaluation of the compressive strength of mortar and concrete

The compressive strength test of the mortar was carried out on the halves of the bars obtained from the flexural tests.

Concrete samples were formed in accordance with the PN-EN 12390-1:2013-03 standard. Three samples were made for each test date. The molded samples in plastic molds. Samples with dimensions of 10 x 10 x 10 cm were made. The samples were disassembled after 1 day and were stored in a climate chamber with water at a temperature of 20±2oC until testing. The strength test was carried out using the testing machine (Fig. 2c). The load was applied perpendicular to the forming direction. The speed of the applied load was 0.6±0.2 MPa/s.

The destructive force is transmitted using metal washers measuring 40 mm x 40 mm and 10 mm thick. The top plate is fitted with holes for guide pins. It is supported by springs. In the center of each plate there is a rectangular plate. They are placed in such a way that when the upper plate is pressed against the lower plate, the surfaces overlap. The halves of the bars, which are obtained after bending, are placed symmetrically on the surface of the plate. The pressure increase per sample should be 15-20 kg/cm2 per second. When crushing the sample, it is important to read the pressure on the pressure gauge.
The measure of compressive strength is nothing else than the ratio of the force indicated by the dynamometer or corresponding to the greatest pressure indicated by the manometer to the surface area. The compressive strength of the mortar is the arithmetic average of the strength of the six halves. If one of the results is 5% greater than the calculated average, the result is incorrect. It should be discarded and the average of the remaining results calculated. The test should be repeated when two results differ from the calculated mean by more than 5%.

LINE 269-271: Can you explain in the manuscript why it is suitable as claimed even with lower crystallinity? – as indicated in the manuscript, this material provided a very reasonable price to quality ratio. Since such CNTs can be bought for ca. $100/kg, they can be considered for large scale applications, in contrast to other price prohibitive materials worth much more.

LINE 274-275: “Surfactants can adsorb on the surface of cement grains and CNTs through electrostatic interactions, increasing the hydrophobicity of the grains.”This is under results. The statement above is not supported by any experimental proof in this work. Appears like a conjecture – citations were added to support this statement.

LINE 278: “…..in none of the cases proper dispersion could be obtained”. What is the proof that proper dispersion was not obtained? - in the case of SP1-SP4 admixtures, regardless of the processing conditions (Table 1), in none of the cases proper dispersion could be obtained. Increase in sonication time/amplitude or the amount of admixture were insufficient to facilitate individualization of MWCNTs. Upon the sonication completion, the nanocarbon content rapidly deposited on the bottom and on the walls of the containers (Fig 4b). These superplastizers were primarily made of acrylic or (carboxylate ether) polymers, which did not have appropriate affinity to MWCNTs to overcome their tendency to agglomerate as a result of van der Waals forces. Figure 4b was added to visualize these results.

a) b)

Figure 4. The MWCNTs dispersion in water facilitated by the concrete admixture: a) PL1 (Sample #18 prepared according to the conditions reported in Table 4), b) SP3 (Sample #18).

LINE 303:Write “mind” not “ming” – corrected.

LINE 309: Delete “consistency” – corrected.

LINE 332-333: "The results show that the difference inconsistency is two levels with respect to the composition without MWCNTs." This sentence lacks clarity. What is the exact meaning of the phrase “difference inconsistency is two levels”? – corrected.

LINE 337: Write “yield” instead of “bring” – corrected.

LINE 343: Delete “the” – corrected.

LINE 350: Write “The shortest time of is 625 minutes was for the sample without MWCNTs (Z0).” Instead of “The shortest time was for the sample without MWCNTs (Z0) is 625 minutes” – corrected.

LINE 357: “C3A” Abbreviation is undefined – corrected.

LINE 457-458: The characteristic point (local minima) of the connection of the two areas is frequently used to conclude about condition of a specimen. Citations are much needed to support this statement - according to this suggestion, we added suitable citations, it is [105] and [106] in line 474 of the corrected manuscript

LINE 490-492: “That is because, MWCNTs’ mechanical characteristics are predestined to enhance the flexural or tensile rather than compressive properties.” Predestined? How? – that is because CNTs tend to collapse under compression whereas they are much more resistant to bending or elongating. The following reference [10.1016/j.carbon.2019.05.036] and relevant changes were made in the text to explain it.

Round 2

Reviewer 1 Report

Comments and Suggestions for Authors

The paper was strongly improved after revision and can be accepted in its present form.

Comments on the Quality of English Language

Minor editing of English language required.

Author Response

The paper was strongly improved after revision and can be accepted in its present form.

Comments on the Quality of English Language: Minor editing of English language required.

Thank you for your sincere review. We addressed your comments carefully and proofread the manuscript. We would very much appreciate it if you could reconsider the revised manuscript.

Reviewer 2 Report

Comments and Suggestions for Authors

Dear authors,

I checked very carefully the revised version of your manuscript. I remarked that you considerably improved your manuscript according to my recommendations and comments.

In order you continue to improve your manuscript, I would also like to recommend two changes and improvements which are related with my questions from the 1st my review report, as it is shown below.

1.In the text related to Figure 2, the authors should give details about the type (model) and manufacturer for the equipment CONTROLS used in bending and compression tests.

2. Regarding the results reported in Table 7, your conclusions cannot be supported only by the results obtained for a single sample from each type of mortar Z0 ... Z4. Please, repeat the flexural tests for other minimum two specimens made of each type of mortar and for each testing time so as finally, you’ll have minimum three results for each mortar and for each time of testing. It is known that it is usually recommended minimum 3 specimens (5 specimens is the best) for any mechanical test. Otherwise, the experimental results are not conclusive. In this way, you’ll be able to report stdev in Table 7.

Comments on the Quality of English Language

Dear authors,

In my opinion, the text is quite well written in English, only small corrections are needed.

Author Response

Dear authors,

I checked very carefully the revised version of your manuscript. I remarked that you considerably improved your manuscript according to my recommendations and comments.

In order you continue to improve your manuscript, I would also like to recommend two changes and improvements which are related with my questions from the 1st my review report, as it is shown below.

Comments on the Quality of English Language: Dear authors, In my opinion, the text is quite well written in English, only small corrections are needed.

Thank you for your sincere review. We addressed your comments carefully and proofread the manuscript. We would very much appreciate if it you could reconsider the revised manuscript.

  1. In the text related to Figure 2, the authors should give details about the type (model) and manufacturer for the equipment CONTROLS used in bending and compression tests.

The details about the used equipment were included (300/15 kN PILOT Automatic Compression-Flexural Cement Testers, model: 65-L1142, CONTROLS company).

  1. Regarding the results reported in Table 7, your conclusions cannot be supported only by the results obtained for a single sample from each type of mortar Z0 ... Z4. Please, repeat the flexural tests for other minimum two specimens made of each type of mortar and for each testing time so as finally, you’ll have minimum three results for each mortar and for each time of testing. It is known that it is usually recommended minimum 3 specimens (5 specimens is the best) for any mechanical test. Otherwise, the experimental results are not conclusive. In this way, you’ll be able to report stdev in Table 7.

For flexural tests – prism-shaped samples with dimensions of 160 mm x 40 mm x 40 mm were made using metal moulds consisting of movable walls that formed three chambers after assembly. From each type of mortar (Z0...Z4), 3 moulds were made containing 3 beams. (Fig. 1). The beams were unmoulded and weighed after 1 day, then placed in a climatic chamber until the moment of testing. Flexural tests were performed for three beams of the same composition, for each composition (Z0, Z1, Z2, Z3, Z4) for each testing time. The results given in Table 7 are averages of 3 results obtained for a specific composition.

Fig. 1. Filled moulds for preparing samples for strength tests in accordance with the PN-EN 1015-11:2001 standard

In the subsection: „2.2.7.1 Evaluation of the flexural strength of mortar” the information about measuring multiple samples was added. In addition, the contents of Table 7 were updated to include the error analysis.

Reviewer 3 Report

Comments and Suggestions for Authors

General Comments

The authors have improved the quality of the manuscript based on earlier comments. The have also made acceptable rebuttals to earlier expressed concerns.

A few persistent errors and suggestions to address them have been highlighted to the authors under specific comments below.

Specific Comments

LINE 37-39: Cross-check and rephrase this sentence. It is not possible to decipher the meaning. It seems some words are missing.

LINE 70: Write “power” instead of “amplitude”.

LINE 72: The phrase “proper crystallinity of the filler” is out of place in the sentence and confusing.  Please rephrase with appropriate wordings to convey your thoughts.

LINE 91: Write “to” instead of “for”.

LINE 136: Is it “power” or “amplitude? Please conform the appropriate word.

LINE 144: Write “dispersions” instead of “dispersiony”.

LINE 305: Delete “[X]”.

LINE 348-350: Sentence is difficult to understand. Please rephrase to improve clarity.

LINE 366: Write “The shortest selling tine of 625…” instead of “The shortest time of is 625…”.

LINE 511: Write “In addition” instead of “What is more”.

Comments on the Quality of English Language

Moderate English editing might be necessary to improve both the fluidity and clarity in understanding the manuscript.

Author Response

Reviewer 3

The authors have improved the quality of the manuscript based on earlier comments. The have also made acceptable rebuttals to earlier expressed concerns.

A few persistent errors and suggestions to address them have been highlighted to the authors under specific comments below.

Moderate English editing might be necessary to improve both the fluidity and clarity in understanding the manuscript.

Thank you for your sincere review. We addressed your comments carefully and proofread the manuscript. We would very much appreciate if it you could reconsider the revised manuscript.

Specific Comments

LINE 37-39: Cross-check and rephrase this sentence. It is not possible to decipher the meaning. It seems some words are missing. Thank you, it was corrected.

LINE 70: Write “power” instead of “amplitude”. Thank you, it was corrected.

LINE 72: The phrase “proper crystallinity of the filler” is out of place in the sentence and confusing.  Please rephrase with appropriate wordings to convey your thoughts. Thank you, it was corrected.

LINE 91: Write “to” instead of “for”. Thank you, it was corrected.

LINE 136: Is it “power” or “amplitude? Please conform the appropriate word. Thank you, it was corrected and it now spells „power”

LINE 144: Write “dispersions” instead of “dispersiony”. Thank you, it was corrected.

LINE 305: Delete “[X]”. Thank you, it was corrected.

LINE 348-350: Sentence is difficult to understand. Please rephrase to improve clarity. Thank you, it was corrected.

LINE 366: Write “The shortest selling tine of 625…” instead of “The shortest time of is 625…”. Thank you. It was corrected to: “The shortest setting times of 625...”.

LINE 511: Write “In addition” instead of “What is more”. Thank you, it was corrected.

Round 3

Reviewer 2 Report

Comments and Suggestions for Authors

Dear authors,

I checked very carefully the 2nd revised version of your manuscript. I remarked that you considerably improved your manuscript according to my recommendations and comments.

I would like to recommend your manuscript for publishing in Materials journal.

Comments on the Quality of English Language

In my opinion, the text is quite well written in English, just minor editing of English language is required.